# Revolutionizing Ischemic Stroke Diagnosis and Treatment: The Promising Role of Neurovascular Unit-Derived Extracellular Vesicles

**DOI:** 10.3390/biom14030378

**Published:** 2024-03-20

**Authors:** Xiangyu Gao, Dan Liu, Kangyi Yue, Zhuoyuan Zhang, Xiaofan Jiang, Peng Luo

**Affiliations:** 1Department of Neurosurgery, Xijing Hospital, Fourth Military Medical University, Xi’an 710032, China; gaoxiangyu@fmmu.edu.cn (X.G.); liudan801124@163.com (D.L.); yuekangyi@fmmu.edu.cn (K.Y.); zzy990826@126.com (Z.Z.); 2School of Life Science, Northwest University, Xi’an 710032, China

**Keywords:** ischemic stroke, extracellular vesicle, exosomes, neurovascular unit, blood–brain barrier, diagnosis, biomarker, treatment, therapeutic measures, microRNA

## Abstract

Ischemic stroke is a fatal and disabling disease worldwide and imposes a significant burden on society. At present, biological markers that can be conveniently measured in body fluids are lacking for the diagnosis of ischemic stroke, and there are no effective treatment methods to improve neurological function after ischemic stroke. Therefore, new ways of diagnosing and treating ischemic stroke are urgently needed. The neurovascular unit, composed of neurons, astrocytes, microglia, and other components, plays a crucial role in the onset and progression of ischemic stroke. Extracellular vesicles are nanoscale lipid bilayer vesicles secreted by various cells. The key role of extracellular vesicles, which can be released by cells in the neurovascular unit and serve as significant facilitators of cellular communication, in ischemic stroke has been extensively documented in recent literature. Here, we highlight the role of neurovascular unit-derived extracellular vesicles in the diagnosis and treatment of ischemic stroke, the current status of extracellular vesicle engineering for ischemic stroke treatment, and the problems encountered in the clinical translation of extracellular vesicle therapies. Extracellular vesicles derived from the neurovascular unit could provide an important contribution to diagnostic and therapeutic tools in the future, and more studies in this area should be carried out.

## 1. Introduction

Stroke is the second most lethal and disabling disease in the world. According to statistics from the World Health Organization, approximately 3 million people worldwide die of stroke every year [1]. Stroke includes hemorrhagic stroke (HS) and ischemic stroke (IS). HS is an intracranial hemorrhage caused by hypertension, aneurysm, and other conditions. IS refers to the necrosis of brain tissue due to ischemia and hypoxia caused by blood circulation disorders of the brain. According to statistics, IS accounts for 87% of all strokes [2]. Typical IS symptoms include sudden severe headache; the unilateral weakness, numbness, or paralysis of the face, arms, or legs; blindness or diplopia in one or both eyes; slurred, incoherent, or difficult-to-understand speech; loss of balance or coordination; and non-upright vertigo [3]. The risk factors of IS can be divided into two categories: non-intervention and intervention. The risk factors associated with non-intervention include age, sex, race, and heredity, which cannot be selected or changed by individuals. The risk factors that can respond to intervention include hypertension, diabetes, hyperlipidemia, smoking, drinking, obesity, lack of exercise, and atrial fibrillation, which can be controlled or reduced by lifestyle improvements or drug treatment. Generally speaking, the greater the number and degree of risk factors, the greater the risk of developing IS [4,5]. There are approximately 12 million new IS patients worldwide every year. IS imposes a heavy burden on society due to its high incidence, high mortality rate, and high disability rate. Therefore, the accurate diagnosis and effective treatment of IS are important.

At present, the diagnosis of IS mainly relies on brain imaging tests, which can show the location and extent of brain injury and help distinguish between IS and HS. The most common brain imaging tests are computed tomography (CT) scans and magnetic resonance imaging (MRI) scans [6,7]. The CT scan is the most common and widely available brain-imaging technique for IS [8]. This approach uses X-rays to create a cross-sectional image of the brain. The advantages of the CT scan are that it is fast, easy, and inexpensive, and it can rule out HS and other causes of stroke-like symptoms, such as tumors or infections. The disadvantages of the CT scan are that it has low sensitivity and specificity for IS, especially in the early stages, and it exposes the patient to radiation. CT scans may also miss small and deep injuries that may not be visible on the image [9]. MRI scanning is a more advanced and detailed brain-imaging technique for IS. It uses a magnetic field and radio waves to produce an image of the brain. The advantages of the MRI scan are that it has high sensitivity and specificity for IS, and it can show small and deep injuries that may not be visible on a CT scan [10]. MRI scanning can also provide information on the age, size, and type of the stroke, and the status of the blood–brain barrier (BBB) and the penumbra (the area of potentially salvageable brain tissue) [11]. The disadvantages of the MRI scan are that it is more expensive, time consuming, and less available than a CT scan, and it may be contraindicated in patients with metal implants, pacemakers, or claustrophobia [12]. These brain-imaging tests are expensive and require large-scale imaging equipment in large hospitals. In addition, although imaging results can reveal the area of ischemia, they cannot provide information on disease progression or patient prognosis. Therefore, identifying body-fluid-based IS biomarkers that can be easily measured and that are highly specific and sensitive is expected to solve this problem. Recently, Tao et al. found that the expression levels of MRPS11 and MRPS12 in the peripheral blood of IS patients were significantly decreased, which meant MRPS11 and MRPS12 were expected to be biomarkers for the diagnosis of IS [13]. Additionally, the concentration of neurofilament light chain (NfL) and glial fibrillary acidic protein (GFAP) in the serum of IS patients correlated with clinical outcomes, which meant that NfL and GFAP were expected to be biomarkers for the prognosis of IS [14]. More studies with large cohorts on the biomarkers of IS are still needed.

IS is a serious condition that requires immediate medical attention. The main goal of treatment is to restore blood flow to the affected part of the brain and prevent further damage. At present, the main methods for treating IS are thrombolysis, endovascular thrombectomy, and antithrombotic therapy. Thrombolysis is the use of drugs that dissolve blood clots that block the cerebral arteries. The most common drug is alteplase, which can be given intravenously within 4.5 h of symptom onset [15,16]. However, this treatment has a risk of bleeding complications, especially in the brain, and is not suitable for everyone [17]. Endovascular thrombectomy is a procedure that involves inserting a catheter through a large artery in the groin or arm and advancing it to the site of the clot in the brain [18,19]. A device, such as a stent retriever or an aspiration catheter, is then used to remove the clot and restore blood flow. This treatment can be performed within 6 h of symptom onset, or up to 24 h in some cases, depending on the imaging findings and the patient’s condition. This treatment has been shown to improve the outcomes and reduce the disability of patients with large vessel occlusion, which is a severe type of IS [20,21]. Antithrombotic therapy is the use of drugs that prevent the formation or growth of blood clots. Antithrombotic therapy includes antiplatelet drugs, such as aspirin, clopidogrel, and ticagrelor, and anticoagulant drugs, such as heparin, warfarin, and dabigatran [22]. Antithrombotic therapy is usually started within 24 to 48 h of symptom onset, unless there is a high risk of bleeding or a need for thrombolysis or thrombectomy. Antithrombotic therapy is continued for the long term to prevent recurrent strokes and other cardiovascular events [23]. These mainstream treatment methods can effectively alleviate the patient’s symptoms in the early stage of IS [24,25], but they do not completely restore neurological function when administered in the later stage, leading to a significant incidence of disability. Therefore, more effective treatments that promote the recovery of neurological function in patients in the later stage are urgently needed.

Extracellular vesicles (EVs) are small particles that are released by cells and can carry different types of molecules, such as proteins, lipids, and nucleic acids. EVs can travel in the body fluids and interact with other cells, either by attaching to their surface or by being taken up inside them. EVs can then transfer their molecules to the recipient cells, influencing their functions and behaviors [26]. EVs have been suggested as a novel promising tool for the diagnosis and treatment of various diseases, especially in the field of neurology. EVs can cross the BBB, which is a major obstacle for delivering drugs and biomarkers to the brain. EVs can also modulate the immune system, the coagulation process, and the development of diseases such as cancer and neurodegeneration [27]. EVs can be used as natural carriers for delivering drugs, genes, or other therapeutic agents to the brain. EVs can be engineered to carry specific cargoes, such as anti-inflammatory agents, antioxidants, neurotrophic factors, or siRNAs, and to target specific receptors or cells in the brain. EVs can also enhance the stability, bioavailability, and efficacy of the drugs, and reduce their toxicity and side effects [28,29]. In addition, EVs can serve as biomarkers for different neurological diseases, such as Alzheimer’s disease (AD), Parkinson’s disease (PD) and multiple sclerosis. EVs can reflect the molecular and cellular changes that occur in the brain and can be detected in minimally invasive procedures, such as blood or cerebrospinal fluid (CSF) samples. EVs can provide information on the diagnosis, prognosis, and response to therapy of these diseases [30]. Additionally, EVs can promote the regeneration and repair of the damaged brain tissue after injury or disease. EVs can stimulate the survival, proliferation, differentiation, and migration of neural stem cells (NSCs), and enhance their integration into the existing neural network. EVs can also modulate the inflammatory and immune responses, and reduce the oxidative stress and apoptosis in the brain [31,32]. Recently, the potential of neuronal EVs in the diagnosis and treatment of neurological diseases has been recognized. Therefore, the applications of neuronal EVs as biomarkers and therapeutic tools in IS have been explored in an increasing number of studies [33,34,35]. This review focused on recent work on the potential role of neuronal EVs in IS.

## 2. Neurovascular Unit

The neurovascular unit (NVU), proposed by scientists in 2001, is composed of several components including neurons, astrocytes, microglia, oligodendrocytes, vascular endothelial cells (ECs), perivascular cells (PCs), vascular smooth muscle cells, the basement membrane, and the extracellular matrix [36]. The tubular arrangement of capillaries within the brain is composed of ECs, while PCs and astrocyte endfeet enclose the perimeter of this tubular structure. Surrounding the outermost part of the tubules is a basement membrane, which consists of an extracellular matrix. Together with neurons, oligodendroglia, and microglia, these components constitute the NVU (Figure 1). In recent years, with the deepening of neuroscience research, the NVU has received increasing attention.

### 2.1. Function of the Neurovascular Unit in Central Nervous System Physiology

The NVU plays an important role in normal physiological processes in the human body, including regulating the BBB and intracranial blood flow [37]. Intracranial capillaries are composed of ECs, and there are tight junctions and adherens junctions between ECs. Tight junctions connect ECs and control the paracellular permeability of the BBB, while adherens junctions regulate cell–cell adhesion. Tight junctions are mainly composed of several important transmembrane proteins, including zonula occludens-1 (ZO-1), occludin, claudin, and junctional adhesion molecules [38]. Adherens junctions are mainly composed of vascular endothelial cadherin [39]. Tight junctions and adherens junctions play an important role in the BBB. Tight junctions can block the free passage of bioactive substances, while adherens junctions play an important role in cell–cell contact and cell development [37]. PCs cover the outer side of capillaries and can regulate blood flow by controlling the diameter of the capillaries. In addition, PCs can also participate in clearing toxic proteins and forming tight junctions [40]. Astrocytes are the most widely distributed cells in the central nervous system. Astrocytes can transmit information between neurons and capillaries through a process known as neurovascular coupling. Microglia are resident immune cells in the central nervous system. Recent studies have found that microglia in the resting state can not only express claudin-5, but also promote ECs to express occludin and ZO-1, thereby maintaining the integrity of the BBB [41,42]. Oligodendrocytes can generate myelin, which can facilitate the efficient transmission of electrical signals and ensure communication between neurons in different areas [43]. To sum up, the NVU plays an important role in maintaining the stability of brain function.

### 2.2. Function of the Neurovascular Unit in Central Nervous System Diseases

The NVU plays an important role in central nervous system diseases, including AD, PD, amyotrophic lateral sclerosis (ALS), Huntington’s disease (HD), and reversible cerebral vasoconstriction syndrome (RCVS). The most typical pathological feature of AD is the deposition of a large number of amyloid β (Aβ) proteins in the brain. NVU dysfunction in patients with AD leads to decreased cerebral blood flow (CBF) and increased BBB permeability, which, in turn, reduces Aβ clearance efficiency and increases Aβ deposition [44]. The main pathological mechanism of PD is the loss of function of dopaminergic neurons in the ventral midbrain. With aging, NVU dysfunction leads to the increased permeability of the BBB, causing the deposition of α-synuclein and other neurotoxic substances, further damaging dopaminergic neurons [45]. Additionally, studies have shown that mice with ALS have lower levels of tight junction proteins, including ZO-1, occludin, and claudin. In the NVU, the loss of tight junction proteins leads to the impairment of BBB function, allowing many harmful substances to enter the brain parenchyma and destroy motor neurons, further promoting the development of ALS [46]. HD was initially considered to be a disease only involving neurons, but recent studies have found that glial cells and ECs in the NVU are also involved in the pathophysiology of the disease [47]. Astrocytes in HD patients can further damage neurons by inhibiting neuronal maturation, increasing neuronal autophagy, and promoting neuroinflammatory responses, thus aggravating the condition of HD [48,49,50]. In addition, HD patients will have pathological phenomena such as capillary leakage and abnormal angiogenesis, leading to impaired BBB function. Toxic substances will accumulate or penetrate the BBB into the brain parenchyma, further damaging neurons and aggravating HD [51]. RCVS is a complex neurovascular syndrome characterized by the sudden onset of severe headache. A clinical study conducted in 2017 showed that 70% of RCVS patients had BBB dysfunction, which meant that the impairment of the BBB may play key roles in the pathophysiology of RCVS [52]. Since the NVU can regulate the function of BBB, the maintenance of NVU homeostasis is expected to be a potential target for the treatment of RCVS. To sum up, the NVU plays an important role in the development of neurodegenerative diseases.

### 2.3. Role of the Neurovascular Unit in IS

After acute IS, NVU dysfunction leads to an increase in BBB permeability and promotes the occurrence of vasogenic brain edema [53]. In addition, PCs and glial cells can further damage the BBB and aggravate brain edema by promoting neuroinflammation [54]. Cells in the NVU protect each other through information exchange after IS. For example, PCs can protect ECs by releasing angiopoietin-1 and glial-cell-derived neurotrophic factor (GDNF), which, in turn, protects the BBB [55]. PCs can also protect neurons and promote axonal regeneration by releasing nerve growth factor and brain-derived neurotrophic factor (BDNF) [56]. Additionally, astrocytes shift from a resting state to an activated state after IS and can be divided into two subtypes (named A1 and A2). A1 astrocytes play a role in promoting neuroinflammation, whereas A2 astrocytes play a role in inhibiting neuroinflammation [57]. Coincidentally, microglia can also be divided into two subtypes (named M1 and M2) after IS. M1 microglia release inflammatory factors such as interleukin (IL)-1, IL-6, and tumor necrosis factor-α (TNF-α) to disrupt the homeostasis of the central nervous system, whereas M2 microglia suppress neuroinflammation [58,59].

Age is an important risk factor for IS and the incidence of IS increases with age [1]. The function of the NVU may become disordered after aging. For example, the dysfunction of aging ECs leads to the impairment of the integrity of the BBB, which is more likely to induce the occurrence of IS. In addition, aging is accompanied by risk factors such as hypertension, diabetes, and hyperlipidemia, which lead to a larger cerebral infarction area and worse neurological recovery [60]. One study showed that older rats suffered more neurological loss after IS than younger rats [61]. Therefore, we should scientifically use aged animal models as much as possible when studying the pathological mechanism, diagnosis, and treatment of IS [62].

## 3. Extracellular Vesicles

### 3.1. Extracellular Vesicles

Extracellular vesicles are nanoscale lipid bilayer vesicles secreted by various cells. The contents of EVs are highly heterogeneous due to differences between recipient or source cells, making it difficult to subclassify EVs based on their contents [63]. In addition, there are currently no specific biological genetic markers to identify EV subtypes [64]. Therefore, the International Society for Extracellular Vesicles (ISEV) recommends the use of “extracellular vesicles” as a generic name for vesicles released by cells [63]. According to the size of the vesicle diameter, EVs can be roughly divided into exosomes, microvesicles, and apoptotic vesicles [27]. Exosomes are small EVs with a diameter of approximately 50–150 nm. Exosome biogenesis occurs through the endocytic endosomal pathway and there are some specific markers on the membrane surface of exosomes, including CD9, CD63, CD81, ALG-2-interacting protein X (ALIX), tumor susceptibility gene 101 (TSG101), Annexins, and heat shock proteins (HSP) 70 and HSP 90 [65]. Microvesicles are EVs with a diameter of 100–1000 nm and are formed by the detachment of the cell membrane after direct outward budding. Similarly, there are also some specific markers on the membrane surface of microvesicles, including phosphatidylserine (PS), selectins, ADP-ribosylation factor 6 (ARF6), CD40, and Rho family members [65]. The diameter of apoptotic vesicles ranges from 100 to 1000 nm and they are the products of programmed cell death [66]. Due to the large heterogeneity in the diameter of apoptotic vesicles, there are few studies on vesicles of this type. Recently, scholars have found that apoptotic vesicles derived from different cells have different biological functions, including inducing inflammation and immune rejection, inducing liver fibrosis, and maintaining the malignant behavior of malignant tumors [67,68,69]. The secretion of EVs was initially thought to be a process by which cells excrete intracellular waste products. However, in recent years, scholars have found that EVs are also responsible for intercellular material exchange and information exchange because EVs can carry a large amount of bioactive substances, including nucleic acids, proteins, and lipids, which can play an important role in normal cellular homeostasis or pathological progression [70].

The bioactive substances in EVs, such as proteins and nucleic acids, can reflect the physiological state of parental cells [71]. In addition, EVs are widespread and abundant in the various bodily fluids of organisms, such as blood, CSF, urine, saliva, breast milk, and semen. Therefore, an increasing number of scholars believe that EVs may become biomarkers of diseases. Tumor cells can secrete many EVs, and many studies have confirmed that EVs can serve as effective markers of tumors [72,73,74,75,76]. The levels of seven miRNAs in the serum EVs of patients with colorectal cancer were found to be significantly increased. After the surgical resection of colorectal tumors, the levels of these seven miRNAs in serum EVs were significantly reduced [77]. Another study confirmed that the levels of miR-375 and miR-1290 in serum EVs could be used to predict the prognosis of prostate cancer patients [78]. In addition to EVs in serum, the levels of miR-21 in CSF EVs of glioblastoma (GBM) patients were significantly increased [79]. In addition to tumors, EVs can also serve as biomarkers for many other diseases, including neurological diseases. The expression levels of miR-1, miR-153, miR-19b-3p, miR-10a, miR-19b, miR-409-3p, let-7g-3p, miR-24, and miR-195 were found to be downregulated in EVs isolated from the CSF of patients with PD [80]. Another study demonstrated elevated levels of miR-34a-5p in EVs isolated from the plasma of patients with PD, and receiver operating characteristic curve (ROC curve) analysis indicated the excellent diagnostic potential of miR-34a-5p for distinguishing PD patients from controls [81]. The levels of miR-132 and miR-212 in brain-derived plasma EVs have good specificity and sensitivity for the diagnosis of AD [82]. In addition, the levels of miR-6165, miR-3679-5p, miR-574-5p, and miR-6760-5p in EVs extracted from CSF can be used to diagnose Moyamoya disease (MMD) [83]. Therefore, EVs have great potential to become effective biomarkers of diseases, including neurological diseases.

The bioactive substances carried by EVs can be internalized by target cells and alter their biological processes. In addition, the immunogenicity of EVs is low, and they can pass through the BBB. Therefore, increasing research is focusing on the therapeutic effect of EVs on many diseases, including neurological diseases. A study showed that the injection of EVs derived from mesenchymal stem cells (MSCs) rich in tyrosine phosphatase-2 into AD mouse models could rescue synapse loss and neurological function decline in AD mice by reducing neuronal apoptosis and inhibiting neuroinflammation [84]. M2 microglia-derived EVs can act on oligodendrocyte precursor cells (OPCs) by delivering miR-23a-5p to promote white matter repair in stroke and demyelinating diseases [85]. In addition, EVs derived from MSCs can effectively protect brain function after stroke through neuroprotection, nerve regeneration, and other effects [86]. Therefore, EVs have broad prospects for the treatment of neurological diseases.

### 3.2. Extracellular Vesicles and IS

Recent studies have shown that EVs play an important role in the pathophysiological process of IS [87,88]. Researchers have found that many different cell-derived EVs have neuroprotective and nerve repair functions. For example, plasma EVs can transfer HSP70 to cells in the cerebral infarction area and inhibit the production of intracellular reactive oxygen species (ROS), thereby alleviating mitochondrial damage and BBB damage [89]. EVs derived from MSCs can promote the recovery of neurological function after IS, but the specific mechanism needs to be further elucidated. Recently, Huang et al. carried out miRNA transcriptome analysis and found that miR-19b-3p, miR-204-3p, miR-125a-5p, miR-672-3p, and miR-667-3p in the EVs derived from MSCs could exert neuroprotective effects by acting on the JAK/STAT, PI3K/Akt, and insulin signaling pathways [90]. Conversely, some cell-derived EVs have been implicated in neurodegeneration. For example, EC-derived EVs could disrupt the BBB [91].

In addition to those roles, EVs can serve as biomarkers for IS. Qi et al. found that the level of miR-124-3p in serum EVs decreased in the early stages of IS and was negatively correlated with serum pro-inflammatory cytokines [92]. Therefore, miR-124-3p was expected to be a diagnostic and prognostic marker for IS. Additionally, EVs can also be used as markers to distinguish different locations of IS. Different regions of the brain have different cellular components. Therefore, the contents in EVs released from different brain regions are different. Through proteomic and transcriptomic analyses, researchers found that the level of miR-15a, miR-100, miR-339, and miR-424 in circulating EVs was lower in cortical–subcortical IS patients than in subcortical IS patients [93]. In addition, EVs can be utilized as markers to distinguish different stages of IS. It was reported that the plasma exosomal miR-21-5p levels in subacute-phase IS and recovery-phase IS were significantly higher than those in other phases IS [94].

### 3.3. Extracellular Vesicles Derived from the NVU

EVs derived from the NVU (NVU-EVs) refer to EVs released from various cell types in the NVU that can exchange information within the NVU or between the NVU and other tissues or organs. The characteristics and classification of NVU-EVs are the same as those of general EVs, but their composition and function may be related to the cell type and state of the NVU.

Recently, some researchers have reported the protective effects of NVU-EVs against neurological diseases. For example, Leggio et al. found that EVs originating from astrocytes in the ventral midbrain and striatum could inhibit neuronal apoptosis by rescuing mitochondrial function in neurons. These data also confirmed that the EVs released by astrocytes in different brain regions have different functions [95]. The interaction between microglia and astrocytes plays a significant role in central sensitization and neuroinflammation, but the mode of glial cell interaction remains less clear. A recent study found that dual immunoglobulin domain-containing cell adhesion molecules (DICAMs) in EVs derived from reactive astrocytes could suppress microglial activation by targeting the mitogen-activated protein kinase (MAPK) signaling pathway and subsequently attenuate neuroinflammation during central sensitization [96]. In a mouse traumatic spinal cord injury model, miR-672-5p in EVs derived from M2 microglia suppressed the AIM2/ASC/caspase-1 signaling pathway in neurons and subsequently promoted the recovery of behavioral function by inhibiting neuronal pyroptosis [97].

In contrast, NVU-EVs also have adverse effects on neurological diseases. Gabrielli et al. found that amyloid-β in microglia-derived EVs promoted the spread of long-term potentiation impairment and synaptic plasticity impairment, ultimately leading to synaptic dysfunction in the early phase of AD [98]. Neuromyelitis optica spectrum disorder (NMOSD) is an inflammatory demyelinating disease. Its pathogenesis is closely related to aquaporin-4 autoantibodies specifically expressed by astrocytes in the central nervous system. It was recently reported that miR-129-2-3p in EVs from astrocytes led to demyelination by targeting the SMAD3 gene in oligodendrocytes and the optic nerve [99].

Scholars have also found that NVU-EVs can be used as biomarkers for the diagnosis of neurological diseases. Pathological α-synuclein can be detected in neuronal EVs from blood plasma samples under normal conditions, and its expression significantly increased in all patients with PD [100]. This finding suggests that pathological α-synuclein in neuronal EVs from plasma samples has the potential to become a blood biomarker of PD. In addition, the measurement of the miR-129-2-3p level in astrocyte-derived EVs from plasma samples might be helpful for the diagnosis of NMOSD because it was found to be significantly increased in patients with NMOSD and was also found to be correlated with disease severity [99]. Myelin and lymphocyte (MAL) protein is specifically expressed by central nervous system ECs, and the expression level of MAL protein can be determined by measuring CD31, CD105, or CD144 expression. Mazzucco et al. used flow cytometry to identify EVs derived from circulating central nervous system ECs in human blood samples and found that an elevated number of EVs from central nervous system ECs is positively correlated with the occurrence of multiple sclerosis (MS) [101].

## 4. Role of NVU-EVs in IS

The NVU plays an important role in the occurrence and development of IS. EVs are important mediators of cellular interactions, and many recent studies have reported the important role of NVU-EVs in IS.

### 4.1. NVU-EVs and IS Diagnosis

As depicted in Figure 2, NVU-EVs might be early biomarkers for IS. In a rat transient ischemic attack (TIA) model, the trend of changes in CSF exosomal miRNA levels was consistent with the trend of changes in plasma exosomal miRNA levels. Li et al. found that the level of miR-122-5p decreased in a 10 min TIA model and that miR-300-3p levels increased in a 5 min TIA model [102]. Luo et al. found that the level of miR-450b-5p increased in a 10 min TIA model compared with control rats [103]. The above studies suggest that circulating/CSF exosomal miR-122-5p, miR-300-3p, and miR-450b-5p could be promising biomarkers for TIA. miR-9 and miR-124 are two brain-specific miRNAs that can be detected in serum EVs. By comparing the levels of serum exosomal miR-9 and miR-124 in patients with IS and volunteers without stroke, researchers concluded that the levels of miR-9 and miR-124 are positively correlated with the National Institutes of Health Stroke Scale (NIHSS) score, serum IL-6 concentration, and infarct volume [104]. This finding suggests that miR-9 and miR-124 in serum EVs could be potential biomarkers for diagnosing IS and evaluating the degree of IS injury. Additionally, the activation of microglia is closely related to poor prognosis in IS, but it is difficult to measure the activation of microglia in vivo [105,106,107]. EVs in plasma are allowed for non-invasive measurement and are cell-specific indicators that might reflect the activation state of microglia [108]. EVs originating from activated microglia contain characteristic proteins (microglial protein TMEM119 and the Toll-like receptor 4 coreceptor CD14) [109]. Roseborough et al. found that the number of TMEM119+/CD14+ EVs in plasma was increased after IS in rats, which indicated the increased microglia activation [109]. This finding demonstrated that EVs derived from activated microglia in the plasma had great potential as prognostic markers for IS. Picciolini et al. found that the level of vascular endothelial growth factor receptor 2 (VEGFR2) in activated microglia-derived EVs in the serum and the level of translocator protein (TSPO) in neuron or microglia-derived EVs in the serum significantly increased after IS [110]. These data meant that serum exosomal VEGFR2 and TSPO might be sensitive biomarkers of IS. Additionally, recent studies have revealed the increased level of phosphatidylserine, Annexin-V, and CD105 and decreased level of CD41a in EC-EVs circulating in the plasma of IS patients [111,112]. Brenna et al. found that prion protein levels were significantly increased in brain-derived EVs in mice with IS and prion protein might contribute to intercellular communication [113]. Zhang et al. found that the number of EC-EVs and level of miR-155 in EC-EVs in the plasma significantly increased in the early and middle stages after IS, were positively correlated with the cerebral ischemic infarction area and NIHSS score, and were related to large-artery atherosclerosis and cardioembolism subtypes [114]. The level of miR-126 in EC-EVs in the serum was found to be decreased at 3 h after cerebral ischemia in rats and to return to normal levels after 24 h [115]. Therefore, serum exosomal miR-126 might be a sensitive biomarker of IS. Together, all of these studies (Table 1) support the potential of bioactive substances from circulating EVs as an early biomarker for IS and a key player in IS development.

### 4.2. NVU-EVs and IS Treatment

Recent studies have found that the bioactive substances carried by NVU-EVs plays an important role in the development of IS and could be beneficial for the treatment of IS.

#### 4.2.1. Neuron-Derived EVs

A study found that EVs could mediate information exchange between neurons and microglia after stroke. Neuron-derived EVs could prevent stressed neurons from being engulfed by microglia by transporting miR-98 to the microglia [116]. Song et al. reported that neuron-derived EVs could downregulate the expression of CXCL1 in astrocytes by transporting miR-181c-3p, thereby alleviating the neuroinflammatory response after IS [117]. In addition, after the injection of neural progenitor cell (NPC)-derived EVs into the tail vein, the inflammatory response in the brains of middle cerebral artery occlusion (MCAO) mice was significantly reduced. Further research has shown that seven miRNAs (let-7g-5p, miR-99a-5p, let-7i-5p, miR-139-5p, miR-98-5p, miR-21-5p, and let-7b-5p) in NPC-derived EVs inhibit inflammation by inhibiting the MAPK pathway in microglia [118]. Furthermore, NPC-derived EVs could reduce the recruitment of inflammatory-related cells by inhibiting the nuclear factor-κB (NF-κB) pathway and enhance BBB integrity by regulating ATP-binding cassette transporter B1 (ABCB1) and matrix metalloproteinase 9 (MMP-9) [119]. Additionally, Xu and his teammates found that EVs derived from NPCs could reduce the infarct volume, neurologic deficit score (NDS), neural apoptosis, and ROS production as well as increase the density of dendrites in MCAO mice, possibly through the regulation of the ROS/Nox2 and BDNF/TrkB pathways by miR-210 [120].

#### 4.2.2. Astrocyte-Derived EVs

Recently, researchers have found that astrocyte-derived EVs could reduce neuronal autophagy and the expression of inflammatory factors, including TNF-α, IL-6, and IL-1β, in neurons after IS, thereby reducing neuronal damage [121]. Furthermore, Pei and his colleagues reported that EVs released by astrocytes could transfer miR-190b to neurons and inhibit IS-induced autophagy and neuronal apoptosis by mediating Atg7 [122]. Additionally, astrocyte-derived EVs could promote recovery after IS by reducing the infarct volume, protecting the function of neuronal tracts, promoting axonal regeneration, and enhancing compound action potential recovery [123]. Recently, a large number of studies proved that berberine had neuroprotective effects on central nervous system diseases, including IS, AD, PD, and schizophrenia [124,125,126,127]. There are even two clinical studies on the efficacy of berberine in the treatment of schizophrenia (NCT02983188, NCT03548155: https://clinicaltrials.gov, accessed on 7 March 2024). Therefore, berberine may be a potential therapeutic option to treat IS. Ding et al. found that astrocytes pre-exposed to oxygen glucose deprivation (OGD) could release EVs after treatment with berberine and EVs could act on neurons and regulate the Rac1 pathway through miR-182-5p, thereby reducing neuronal damage and attenuating the neuroinflammatory response [128]. Hira et al. found that EVs from astrocytes subjected to OGD and treated with a semaphorin 3A inhibitor could promote axonal outgrowth and nerve function recovery after IS through prostaglandin D2 synthase [129].

#### 4.2.3. Microglia-Derived EVs

EVs from microglia pre-exposed to OGD were found to reduce neuronal damage and promote tube formation and angiogenesis in ECs through the regulation of the Smad2/3 pathway in both neurons and ECs by transforming growth factor (TGF)-β1 in microglia-EVs. Additionally, TGF-β1 in microglia-derived EVs could promote the M2 polarization of resident microglia in the ischemic area [130]. Li et al. found that M2 microglia-derived EVs could regulate Olig3 in OPCs through miR-23a-5p after IS, thereby increasing the proliferation, survival, and differentiation of OPCs and promoting brain white matter repair [85]. Additionally, miR-124 in M2 microglia could attenuate glial scar formation by regulating signal transducer and activator of transcription (STAT) 3 in astrocytes. miR-124 not only reduced the expression of GFAP and inhibited the proliferation of astrocytes, but also promoted the conversion of astrocytes into neural precursor cells by increasing the expression of Sox2 and decreasing the expression of Notch 1 [131]. Interestingly, miR-124 in EVs originating from M2 microglia could also promote IS recovery by promoting the proliferation and differentiation of NSCs in MCAO model mice. Adaptor-associated protein kinase 1 (AAK1)/Notch is an important target of miR-124 in NSCs [132]. Furthermore, M2 microglia-derived EVs could transfer miR-137 to neurons, and miR-137 could inhibit neuronal apoptosis by targeting Notch 1 [133]. Another study found that EVs from microglia pre-exposed to OGD could not only inhibit neuroinflammation, AQP4-mediated depolarization, brain edema, and astrogliosis, but also promote CSF flow, cerebral blood circulation, and neurological recovery [134]. However, the mechanism needs to be further clarified. Notably, the contents of EVs derived from microglia have both protective and destructive effects. Xie et al. reported that miR-424-5p was heavily enriched in EVs from ischemia-preconditioned microglia and could be transferred to ECs, and aggravated OGD-induced decreases in EC integrity and viability and loss of vascular formation by mediating the FGF2/STAT3 pathway [135].

#### 4.2.4. Endothelial Cell-Derived EVs

Xiao et al. found that EVs derived from ECs could inhibit OGD-induced neuronal apoptosis, promote the migration and invasion of SH-SY5Y cells, and reduce the infarct volume in the brains of MCAO rats [136]. Additionally, after IS in diabetic mice, miR-126 in EC-EVs could not only promote neurological functional recovery and increase the myelin density, axon density, arterial diameter, and vascular density, but also induce M2 macrophage polarization in the infarct area [137]. Similarly, miR-126 in EVs derived from endothelial progenitor cells (EPCs) could reduce the infarct size, increase CBF and cerebral microvascular density (MVD), and promote angiogenesis, neurogenesis, and neurological functional recovery after IS in diabetic and healthy mice, possibly via the regulation of the ROS/Nox2 and BDNF/TrkB pathways [34,120,138,139]. Furthermore, EC-EVs could promote IS recovery in MCAO model rats by increasing NPC migration and proliferation as well as reducing NPC apoptosis [140]. Gao and his colleagues found that EC-EVs could promote neurological functional recovery in MCAO model mice by mediating the inflammatory response in astrocytes. miR-155-5p in EC-EVs could repress the inflammatory response of astrocytes by regulating the c-Fos/AP-1 pathway [35]. Furthermore, miR-210 in EVs originating from EPCs could reduce apoptosis and ROS production and increase the viability of OGD-injured neurons. The mechanism of action of miR-210 involved the inhibition of neuronal apoptosis through the BDNF/TrkB pathway and the inhibition of ROS production through the Nox2/Nox4 pathway [141]. Li et al. revealed that miR-19a, miR-21, and miR-146a in EC-EVs could repress the expression of prothrombotic and BBB-leakage-related proteins, including Toll-like receptor (TLR) 4, intercellular adhesion molecule-1 (ICAM-1), PAI-1, TF, and NF-κB, in surrounding ECs. Therefore, EC-EVs could promote neurological recovery in MCAO rats by increasing CBF and reducing BBB leakage [142]. Notably, EVs released by ECs under different physiological conditions have diametrically opposite effects [143]. EVs released by ECs under normal conditions (n-EC-EVs) could promote the proliferation of astrocytes by mediating the PI3K/Akt pathway, while EVs released by ECs after OGD (OGD-EC-EVs) had the opposite effect. Additionally, the n-EC-EVs promoted while OGD-EC-EVs inhibited the expression of GFAP in astrocytes, CBF and neurological functional recovery. Moreover, the n-EC-EVs reduced while OGD-EC-EVs increased the apoptosis of astrocytes, the infarct size, and BBB disruption.

#### 4.2.5. Perivascular Cell-Derived EVs

Seifali et al. reported that EVs derived from PCs could inhibit neuronal apoptosis and increase sensorimotor function and the number of microtubule-associated protein 2 (MAP2)+ cells in the infarct boundary area in MCAO model rats [144].

All of the abovementioned literature (Table 2) proved that NVU-EVs played an important role in the development of IS and could be an effective treatment for IS.

## 5. Discussion

The occurrence and severity of IS can be reflected in the expression level of NVU-derived blood EVs, so NVU-derived EVs are expected to become blood biomarkers for the diagnosis and prognosis of IS. In addition, after IS, NVU-derived EVs can transmit the bioactive substances they carry to brain cells, regulate cell function, and promote neural function recovery. In summary, we have clarified the important role of NVU-derived EVs in the diagnosis and treatment of IS in the previous section.

IS is a condition where the blood supply to a part of the brain is interrupted, causing brain tissue damage and neurological deficits. Biomarkers are substances that can be measured in the blood or other body fluids to indicate the presence or severity of a disease. Current biomarkers for IS include GFAP, IL-1β, MMP-9, and miRNAs. GFAP is a protein that is found in astrocytes, a type of glial cell that supports and protects neurons in the brain. GFAP levels increase in the blood after IS, reflecting the extent of astrocyte damage and brain injury [145]. IL-1β is a cytokine that is involved in inflammation and immune responses. IL-1β levels rise in the blood after IS, indicating the activation of inflammatory pathways and the recruitment of immune cells to the site of injury [146]. MMP-9 is an enzyme that degrades the extracellular matrix, a network of proteins and molecules that surrounds and supports cells. MMP-9 levels increase in the blood after IS, reflecting the breakdown of the BBB and the invasion of inflammatory cells into the brain tissue [147]. miRNAs are small molecules that regulate gene expression by binding to messenger RNAs and inhibiting their translation into proteins. miRNAs can act as biomarkers for IS by modulating the expression of genes involved in neuronal survival, inflammation, and vascular function [147]. Some examples of miRNAs that are altered in IS are miR-320b and miR-320d, which are upregulated in the blood and may target genes related to oxidative stress and apoptosis [146]. Compared to traditional biological markers, EVs have lipid bilayers that can protect the bioactive substances inside. In addition, NVU-derived EVs can reflect the physiological and pathological status of the brain cells that produce them. Therefore, NVU-derived EVs can serve as biomarkers for IS. However, there are also some challenges and limitations in using NVU-derived EVs as biomarkers for IS [148]. For instance, the isolation and detection of EVs are not standardized and require sophisticated techniques and equipment. The sources and heterogeneity of EVs may also affect their specificity and sensitivity as biomarkers. The mechanisms and functions of EVs in IS are not fully understood and need further investigation. Therefore, NVU-derived EVs may have potential as biomarkers for diagnosing IS, but more research is needed to validate their clinical utility and to overcome the technical and biological challenges.

The current treatments for IS aim to restore blood flow, reduce inflammation, and prevent complications. At present, the most commonly used treatments for IS include thrombolytic therapy and neuroprotective agents [15]. However, these treatments have significant limitations. Thrombolytic therapy has a short time window, and many IS patients do not have the opportunity of thrombolytic therapy [149]. Traditional neuroprotective agents are drugs that protect the brain cells from the damage caused by ischemia and inflammation. Neuroprotective agents include antioxidants, anti-inflammatory agents, calcium channel blockers, and sodium channel blockers, which have large molecular weight and poor penetration ability, making it difficult to target the brain through the BBB [150]. Therefore, it is important to develop more effective treatments. EVs have great prospects for the treatment of IS, so researchers began to modify EVs to enhance their stability, targeting, and efficacy in IS treatment. EV engineering can be achieved by modifying donor cells, the EVs themselves, or EV cargo. Donor cell modification can be performed by genetic manipulation, such as overexpressing or knocking down specific genes, or by environmental stimulation, such as hypoxia, oxidative stress, or pharmacological agents. For example, MSCs can be engineered to overexpress BDNF, which can increase the BDNF content and neuroprotective effect of MSC-derived EVs in IS [151]. EV modification can be performed by physical, chemical, or biological methods, such as coating, conjugation, or fusion. For example, EVs can be coated with polyethylene glycol (PEG) to increase their stability and circulation time or conjugated with targeting ligands, such as peptides, antibodies, or aptamers, to enhance their specificity and uptake by ischemic brain cells [152]. EV cargo modification can be performed by loading EVs with exogenous molecules, such as drugs, genes, or nanoparticles, or by enriching EVs with endogenous molecules, such as miRNAs, proteins, or lipids. For example, EVs can be loaded with curcumin, a natural anti-inflammatory and antioxidant compound, which can increase the therapeutic effect of EVs in IS [153]. EV engineering methods can also be combined to achieve synergistic effects. For example, EVs can be loaded with magnetic nanoparticles and conjugated with targeting ligands, which can enable the magnetic guidance and enhanced delivery of EVs to the ischemic brain [154].

EVs have shown great potential for IS treatment, but there are still some challenges and limitations that need to be addressed before they can be applied in clinical settings. Some of the challenges and perspectives are as follows: First, the biodistribution of EVs needs to be urgently elucidated to determine their biosafety to allow the clinical application of EVs. This may rely on the use of molecular probes with high labeling efficiency and tracking ability. Currently, the most commonly used commercial EV trackers have a robust quenching effect in aggregates and influence the particle size of EVs, which results in poor tracking ability [155]. The tracking technology of EVs is crucial for understanding their biological behavior in different environments and promoting the clinical translation of EV therapy. The development of powerful molecular imaging technology will further reveal the biogenesis, secretion, transportation, pathophysiological functions, and delivery potential of EVs, even to the precise biological behavior of individual EVs. Therefore, the development of EV tracing technology has profound significance. In bioimaging, aggregation-induced emission (AIE) fluorescent compounds and their nanoparticles have the advantages of high brightness, low background noise, uniform light stability, good biocompatibility, better tissue penetration ability, and higher spatial resolution [156,157,158,159]. Recently, Gao et al. created a new EV tracker with AIE properties that had good labeling efficiency and photostability and successfully tracked EVs in real time in MCAO model mice [35]. Therefore, new EV trackers with AIE properties are expected to overcome the bottleneck of EV tracing. Second, the standardization and optimization of EV isolation, which are currently heterogeneous and inconsistent among different studies, may affect the quality of EVs and the reproducibility of their effects. The low density, small size, and wide distribution of EVs in the complex bodily fluid environment pose a considerable challenge for researchers to obtain high-purity EVs. Currently, the methods for isolating EVs are diverse, including ultracentrifugation, size exclusion chromatography, ultrafiltration, field flow fractionation, and precipitation [27]. Therefore, the identification of a unified and efficient isolation method is conducive to the uniformity of EV research and the large-scale production of EVs. Third, the molecular mechanisms of how EVs exert their effects are not fully understood, and they may vary depending on the type, origin, and composition of EVs, as well as the type and state of the recipient cells. Therefore, more research is needed to explore and verify the interaction and integration of EVs with various cell types, molecules, and pathways. This can be achieved by using advanced techniques such as genomic, proteomic, and metabolomic analyses, as well as imaging and functional assays, to characterize the biogenesis, release, uptake, and cargo of EVs, and to monitor their effects on the recipient cells. For example, Yue et al. found that EC-EVs could act on neurons and inhibit neuronal apoptosis after IS by carrying miR-1290 [160]. Interestingly, Gao et al. found that EC-EVs could act on astrocytes and inhibit astrocyte inflammation after IS by carrying miR-155-5p [35].

## 6. Conclusions

In conclusion, this review focused on the diagnostic and therapeutic role of NVU-EVs in IS. The contents carried by EVs represent the state of the parent cells, so the contents of NVU-EVs can better reflect the state of brain cells after IS. The lipid bilayer of EVs could play a protective role for the bioactive substances they carry, so these bioactive substances are not easily degraded. Compared with bioactive substances in blood, bioactive substances carried by NVU-EVs are more likely to be the diagnostic and prognostic markers of IS. Additionally, accumulating evidence has proven that various NVU-EVs have neuroprotective functions after IS, and revealing the specific mechanisms of NVU-EVs could provide new strategies for brain protection. At present, although there are still difficulties in the clinical translation of NVU-EVs, these vesicles could provide an important contribution to diagnostic and therapeutic tools in the future, and more studies on NVU-EVs should be carried out.

## Figures and Tables

**Figure 1 biomolecules-14-00378-f001:**
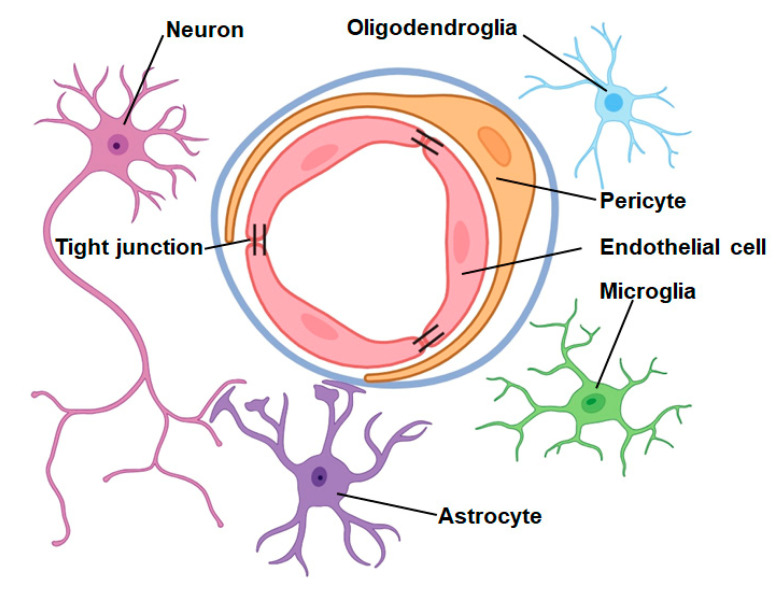
The structure of the neurovascular unit. The ECs form the tubular structure of the capillaries in the brain and ECs are connected by tight junctions. PCs and astrocyte endfeet surround the perimeter of the tubular structure. The basement membrane formed by the extracellular matrix surrounds the outermost part of the tubular structure. These structures, plus neurons, oligodendroglia, and microglia, make up the NVU.

**Figure 2 biomolecules-14-00378-f002:**
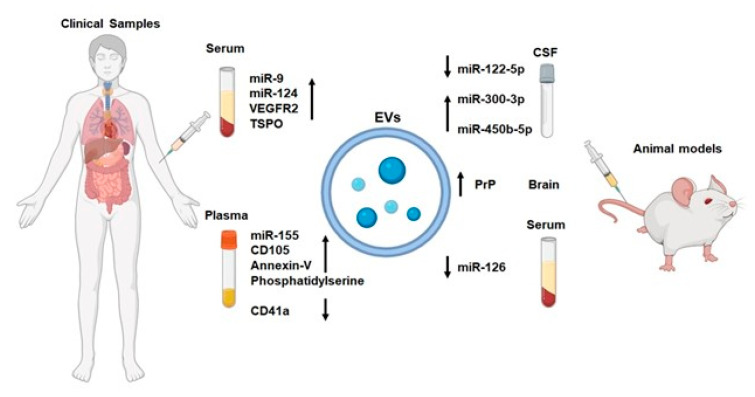
Diagnostic markers carried within NVU-component-derived EVs.

**Table 1 biomolecules-14-00378-t001:** Biomarkers carried by NVU-derived EVs for IS.

Source	Content	Models	Expression in IS	Outcome	Reference
CSF	miR-122-5p	Rats	downregulation	TIA biomarkers	[102]
CSF	miR-300-3p	Rats	upregulation	TIA biomarkers	[102]
CSF	miR-450b-5p	Rats	upregulation	a high diagnostic value and may become a therapeutic target for rat TIA	[103]
Serum	miR-9	Human	upregulation	NIHSS scores, serum IL-6 concentration, infarct volume	[104]
Serum	miR-124	Human	upregulation	NIHSS scores, serum IL-6 concentration, infarct volume	[104]
Plasma, activated microglia-EVs	NA	Rats	upregulation	worse neurological and cognitive outcomes	[109]
Serum, activated microglia-EVs	VEGFR2	Human	upregulation	IS biomarkers	[110]
Serum, neuron/microglia-EVs	TSPO	Human	upregulation	IS biomarkers	[110]
Plasma, EC-EVs	phosphatidylserine	Human	upregulation	IS biomarkers	[111]
Plasma, EC-EVs	CD105	Human	upregulation	IS biomarkers	[111]
Plasma, EC-EVs	Annexin-V	Human	upregulation	IS biomarkers	[112]
Plasma, EC-EVs	CD41a	Human	downregulation	IS biomarkers	[111]
Brain	PrP	Mice	upregulation	intercellular communication at early stages after stroke	[113]
Plasma, EC-EVs	miR-155	Human	upregulation	infarct volume, NIHSS scores, large artery atherosclerosis, cardioembolism subtypes	[114]
Serum, EC-EVs	miR-126	Rats	downregulation	sensitive marker for IS	[115]

IS, ischemic stroke; CSF, cerebrospinal fluid; TIA, transient ischemic attack; NIHSS, National Institutes of Health Stroke Scale; IL, interleukin; EVs, extracellular vesicles; VEGFR2, vascular endothelial growth factor receptor 2; TSPO, translocator protein; EC, endothelial cell; PrP, prion protein.

**Table 2 biomolecules-14-00378-t002:** EVs-based treatment for IS.

Source	Content	Model	Effect	Mechanism	Reference
Neuron	miR-98	tMCAO: rat/mouse OGD: neuron/microglia	Inhibit the microglial phagocytosis of neuron	PAFR	[116]
Neuron	miR-181c-3p	MCAO: rat OGD: neuron/astrocyte	Inhibit neuroinflammation	CXCL1	[117]
NPC	let-7g-5p miR-99a-5p let-7i-5p miR-139-5p miR-98-5p miR-21-5p let-7b-5p	MCAO: mouse	Inhibit neuroinflammation	MAPK pathway	[118]
NPC	NA	tMCAO: mouse OGD: EC	Enhance BBB integrity Attenuate inflammatory cell recruitment	ABCB1/MMP-9 NF-κB pathway	[119]
NPC	miR-210	MCAO: mouse OGD: neuron	Reduce infarct volume, NDS, neural apoptosis and ROS production Promote the spine density of dendrites	ROS/Nox2 pathways BDNF/TrkB pathways	[120]
Astrocyte	NA	MCAO: mouse OGD: neuron	Inhibit neurons apoptosis	autophagy	[121]
Astrocyte	miR-190b	OGD: neuron	Inhibit autophagy and neurons apoptosis	Atg7	[122]
Astrocyte	NA	MCAO: rat	Reduce the infarct volume Protect the function of the neuronal tracts Promote axonal regeneration Enhance compound action potential recovery	NA	[123]
Astrocyte	miR-182-5p	MCAO: mouse OGD: neuron	Reduce neuronal injury Inhibit neuroinflammation	Rac1 pathway	[128]
Astrocyte	NA	MCAO: rat OGD: neuron	Promote axonal outgrowth	prostaglandin D2 synthase	[129]
Microglia	TGF-β1	OGD: neuron OGD: EC	Stimulate both angiogenesis and tube formation Reduce neuronal injury	Smad2/3 pathway	[130]
Microglia	miR-23a-5p	tMCAO: mouse OGD: OPC	Reduce brain atrophy volume Promote functional recovery Promote oligodendrogenesis and white matter repair Increase OPC proliferation, survival, and differentiation	Olig3	[85]
Microglia	miR-124	MCAO: mouse OGD: astrocyte	Attenuate glial scar formation	STAT3 pathway glial fibrillary acidic protein Notch 1/Sox2	[131]
Microglia	miR-124	tMCAO: mouse	Promote proliferation and differentiation of NSCs	AAK1/Notch	[132]
Microglia	miR-137	MCAO: mouse OGD: neuron	Inhibit neuronal apoptosis	Notch 1	[133]
Microglia	NA	MCAO: mouse OGD: microglia/astrocyte	Reduce poststroke inflammation, astrogliosis, AQP4 depolarization Promete CSF flow	NA	[134]
EC	NA	MCAO: rat OGD: neuron/EC	Suppress neuronal apoptosis Promote migration and invasion of neuron	NA	[136]
EC	miR-126	photothrombotic stroke model: mouse	Promote neurological functional recovery Improve myelin density, axon density, arterial diameter, and vascular density Induce M2 macrophage polarization in the infarct boundary zone	NA	[137]
EC	NA	MCAO: rat Cell scratch wound: NPC	Promote NPC proliferation and migration Reduce apoptosis of NPCs	NA	[140]
EC	miR-155-5p	MCAO: mouse OGD: astrocyte	Inhibit the inflammatory response of astrocytes	c-Fos/AP-1 pathway	[35]
EC	NA	MCAO: mouse	Promote proliferation of astrocytes Increase the expression of GFAP Inhibit apoptosis of astrocytes Reduce infarct size and BBB disruption Increase CBF and neurological functional recovery	PI3K/Akt pathway	[143]
EC	miR-19a, miR-21, miR-146a	tMCAO: rat	Reduce prothrombotic and BBB leakage proteins	TLR4 ICAM-1, PAI-1 TF ZO1 NF-κB	[142]
EPC	NA	MCAO: rat	Inhibit cell apoptosis Promote angiogenesis	Wnt3a p-GSK-3β CD31 VEGF	[34]
EPC	miR-126	MCAO: mouse with diabetes	Reduce infarct size Increase CBF and MVD Promote angiogenesis, neurogenesis, and neurological functional recovery	NA	[138]
EPC	miR-126	MCAO: mouse OGD: neuron	Reduce NDS, infarct size, and cell apoptosis rate Increase MVD	BDNF/TrkB/Akt signaling pathway	[139]
EPC	miR-126	MCAO: mouse OGD: neuron	Reduce infarct volume, NDS, neural apoptosis, and ROS production Promote the spine density of dendrites	ROS/Nox2 pathways BDNF/TrkB pathways	[120]
EPC	miR-210	OGD: neuron	Reduce neuronal apoptosis and ROS production Promote neuronal viability	BDNF/TrkB pathways Nox2/Nox4 pathways	[141]
PC	NA	MCAO: rat	Reduce neuronal apoptosis Promote the sensorimotor function	NA	[144]

tMCAO, transient middle cerebral artery occlusion; PAFR, platelet-activating factor receptor; OGD, oxygen glucose deprivation; CXCL1, C-X-C motif chemokine ligand 1; NPC, neural progenitor cell; NA, unknown; MAPK, mitogen-activated protein kinases; BBB, blood–brain barrier; EC, endothelial cell; ABCB1, ATP-binding cassette transporter B1; MMP, matrix metalloproteinase; NF-κB, nuclear factor-κB; NDS, neurologic deficit score; ROS, reactive oxygen species; BDNF, brain-derived neurotrophic factor; OPCs, oligodendrocyte precursor cells; STAT, signal transducer and activator of transcription; NSC, neural stem cell; AAK1, adaptor-associated protein kinase 1; CSF, cerebrospinal fluid; GFAP, glial fibrillary acidic protein; CBF, cerebral blood flow; TLR, Toll-like receptor; ICAM-1, intercellular adhesion molecule-1; ZO-1, zonula occludens-1; EPC, endothelial progenitor cell; VEGF, vascular endothelial growth factor; MVD, microvascular density; PC, perivascular cell.

## Data Availability

Not applicable.

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
