# Peer review of "Revolutionizing Ischemic Stroke Diagnosis and Treatment: The Promising Role of Neurovascular Unit-Derived Extracellular Vesicles"

_biomolecules, 2024, doi:10.3390/biom14030378_

Round 1

Reviewer 1 Report

Comments and Suggestions for Authors

I appreciate the opportunity provided by the editor to review this intriguing manuscript. The current review serves as a comprehensive exploration of the role of extracellular vesicles (EVs) in diagnosing ischemic stroke. I have a few comments to enhance the manuscript:

1. I strongly recommend that the authors consider conducting a meta-analysis. It appears that there is a substantial number of studies available to assess the accuracy of EVs in diagnosing ischemic stroke. A meta-analysis would significantly contribute to understanding the sensitivity of these biomarkers.

2. I suggest dedicating a paragraph to a more in-depth explanation of the functional implications of EVs in ischemic stroke. Expanding on how EVs may be linked to medium to long-term functional outcomes would provide valuable insights for readers.

Comments on the Quality of English Language

The manuscript is well written. 

Reviewer 2 Report

Comments and Suggestions for Authors

Reviewer comments and suggestions

The authors of this study emphasize new ways of diagnosing and treating stroke. Hence, they have discussed how the neurovascular unit, composed of neurons, astrocytes, microglia, and other components, plays a crucial role in the onset and progression of stroke. Additionally, they discussed the key role of extracellular vesicles, which can be released by cells in the neurovascular unit and serve as significant architects of cellular communication, in stroke. Finally, in this review, the authors comprehensively explained extracellular vesicles in the diagnosis and treatment of stroke and the problems encountered in the clinical translation of extracellular vesicle therapies.

Overall, the manuscript is well written. I have listed the concerns and comments that needed to be explained or modified.

  1. Line 32-33 It would be nice if the authors could mention a few biomarkers that were previously studied
  2. Line 68-69 The authors can also mention other glial cells
  3. Line 84-85 It would be nice if the authors could add up more studies on this related topic
  4. Line 123, the authors suggested many studies but cited a single reference, please modify by citing a few more references there
  5. Line 139, 148, and many more, the authors need to use abbreviations if they use the full form at once at the beginning
  6. Line 217-218 How the sentence was connected with the above lines. please explain
  7. Line 242 This section should also go into the subsection of the above.
  8. Line 268-271 Please add up human clinical studies if the topic is available in the literature and cite them
  9. Comments for table Please add up the legend part in the table for easy following for the reader of MS
  10. All references should be modified based on MDPI journal guidelines

Reviewer 3 Report

Comments and Suggestions for Authors

The review aims to explore the significance of extracellular vesicles (EVs) released by the neurovascular unit in stroke diagnosis and treatment. While the topic holds considerable interest and accumulating evidence emphasizes the pivotal role of EVs in post-stroke mechanisms, I believe the current review requires substantial revisions before it can be considered for publication.

Overall, the manuscript lacks readability and reads more like a list of bullet points than a comprehensive review. The description appears excessively schematic.

- The section discussing the "Neurovascular Unit" appears overly simplified. I recommend rephrasing it for better depth. Additionally, while numerous pathologies are mentioned, none are thoroughly explored. If the focus is on stroke, it might be beneficial to limit citations of various neurodegenerative disorders. However, if these references serve to elucidate the role of the Neurovascular Unit, expanding this section could be advantageous.

The focus on the NVU distiguishes this review from other reported manuscript, it should be throughly described.

- I disagree with the section dedicated to EVs; the division into exosomes and microvesicles seems outdated. I suggest providing a more comprehensive explanation and description of the intricate nature of EVs. It would be beneficial to incorporate discussions and references that highlight the diverse and heterogeneous nature of EVs. The description of EVs is oversimplified.

- Authors seem to focus mainly on the role of miRNA derived from NVU-EVs, but many other molecules are enclosed and trasported by EVs and attention should be dedicated to other molecules including proteins adn lipids.

- Bibliography is missing several papers on EVs in stroke. Just some examples: Picciolini et al 2023 IJMS; Hirsch et al. Transl. Stroke Res. 2023; Gualerzi et al. Biology 2021. Consider substantial revision. Otherwise, consider to move the focus of the the review only on miRNA from NVU-EVs.

- No discussion and conclusion session is present. I cannot appreciate the take home message of the authors. Reviews are more than a summary of literature, they guide the reader in the comprehension of a complex field and authors should lead to some conclusions and perspectives.

Again, the perspective section is a list , but some of the challenges actually might need further discussion instead of being simply listed.

Minor:

- revise abbreviations, for example BBB abbreviation is explained in line 145 but it first appears in line 56

Comments on the Quality of English Language

English language and grammar revision would be beneficial to improve readability

Reviewer 4 Report

Comments and Suggestions for Authors

There is a pressing need for innovative approaches in diagnosing and addressing strokes. The neurovascular unit, comprising neurons, astrocytes, microglia, and other elements, plays a pivotal role in initiating and advancing stroke. Recent literature extensively documents the crucial involvement of extracellular vesicles, released by cells in the neurovascular unit, as significant facilitators of cellular communication during strokes. This review aims to outline the contribution of extracellular vesicles to both the diagnosis and treatment of strokes. It will discuss the present state of extracellular vesicle engineering for stroke treatment and shed light on the challenges encountered in the clinical application of extracellular vesicle therapies. From this prospective, this review is of interest. However, given the role of aging as a major risk factor of stroke, the authors shall add a section on the importance of aged models for stroke (see, doi: 10.1080/17460441.2019.1573984)

Comments on the Quality of English Language

The review is certainly of interest to the scholars of stroke research and maybe clinicians

Round 2

Reviewer 1 Report

Comments and Suggestions for Authors

 I thank the authors for their reply. I do not have further comments.

Author Response

Thank you for your comments.

Reviewer 3 Report

Comments and Suggestions for Authors

The Authors have significantly improved the manuscript that has now ameliorated in readability, besides, I now appreciate the proposed overview and discussion of current literature in the field.

Author Response

Thank you for your comments.

Reviewer 4 Report

Comments and Suggestions for Authors

The authors have adequately addressed my concerns

Comments on the Quality of English Language

The authors have adequately addressed my concerns

Author Response

Thank you for your comments.